# Proximate Composition and Bioactive Compounds of Cocoa Bean Shells as a By-Product from Cocoa Industries in Indonesia

**DOI:** 10.3390/foods12173316

**Published:** 2023-09-03

**Authors:** Mohamad Djali, Kimberly Santasa, Rossi Indiarto, Edy Subroto, Fetriyuna Fetriyuna, Elazmanawati Lembong

**Affiliations:** Department of Food Industrial Technology, Faculty of Agro-Industrial Technology, Universitas Padjadjaran, Sumedang 45363, Indonesia; kimberly19003@mail.unpad.ac.id (K.S.); rossi.indiarto@unpad.ac.id (R.I.); edy.subroto@unpad.ac.id (E.S.); fetriyuna@unpad.ac.id (F.F.); elazmanawati.lembong@unpad.ac.id (E.L.)

**Keywords:** cocoa bean shells, criollo, forastero, roasting, fermentation

## Abstract

Cocoa bean shell (CBS) is a by-product from cocoa processing which is abundant in Indonesia, one of the largest cocoa-producing countries. It has a great potential for being processed into food ingredients due to its comparable composition to cocoa nibs. The present study was conducted to identify the proximate composition and bioactive compounds in CBS produced at several cocoa industries in Indonesia utilizing different cocoa varieties (Criollo and Forastero) and processing techniques (fermented, non-fermented, pulp washing, and drying), which remain unknown. The results showed that the CBS derived from roasted Criollo cocoa pods in the Kendeng Lembu cocoa industry had ash and protein content of about 8.21% and 18.79%, respectively, which was higher than other industries. Additionally, the concentration of bioactive substances was higher here than it was elsewhere. This included total phenolic (136.2 mg GAE g^−1^) and theobromine (22.50 mg g^−1^). The lowest ash and protein concentration found in CBS was from Forastero cocoa pods, non-fermented like Sulawesi cocoa. These values were 6.48% and 15.70%, respectively. The concentration of theobromine (15.40 mg g^−1^) was also lower compared to other industries.

## 1. Introduction

Cocoa (*Theobroma cacao* L.) is a plantation crop that provides income to more than 4.5 million families globally [1]. Approximately 89% of the production is supplied by Ghana, Ivory Coast, Nigeria, Cameroon, New Guinea, Brazil, Indonesia, and Ecuador [2]. Indonesia is the third-largest cocoa-producing country in the world, contributing about 15% of global cocoa production [3]. Cocoa has a bright prospect in this country due to its high selling price and marketability, and is one of the income sources. Cocoa has been cultivated in several locations in Indonesia, mainly Java, Sumatra, and Sulawesi. According to the Indonesian Ministry of Agriculture [4], the production of dried cocoa bean in Indonesia reached 728,046 tons in 2021.

During the processing of chocolate products, cocoa bean generates by-products in the form of cocoa bean shell (CBS), cocoa pod husks, and cocoa pulp. CBS is the outer layer attached to the cocoa bean and accounts for approximately 14–15% of the total bean weight [5]. To obtain cocoa bean as a raw material for chocolate products, a roasting process is carried out. During roasting, the moisture content of the bean decreases by about 2%, resulting in the development of the chocolate flavor [6]. The bean also becomes darker and more brittle, making it easier to separate the nib from the shell.

In recent times, CBS has garnered more attention among cocoa researchers for functional food purposes because of its bioactive compounds that are not significantly different from cocoa nibs. It was reported that the bioactive compounds and lipid profile of CBS are similar to that of cocoa butter, besides its chocolate flavor and color [7]. CBS has several beneficial compositions, such as dietary fiber (61.18–65.58 g 100 g^−1^), epicatechin (4.56–6.33 mg g^−1^), catechin (2.11–4.56 mg g^−1^), methylxanthines theobromine (7.12 to 12.77 mg g^−1^), caffeine (4.02 to 6.13 mg g^−1^), polyphenol (6.10–42.97 mg GAE g^−1^), fatty acid (oleic, stearic and palmitic acids), and more; consequently, it serves as a source of nutrition for functional food ingredients, mainly baked goods and beverages [8,9]. CBS has anti-inflammatory and antitumoral activity, which can reduce the risk of several diseases, including inflammatory gut diseases and colorectal cancer [10]. Cocoa industries in Indonesia use different cocoa varieties and processing methods. Two commercial cocoa varieties used for processing in Indonesia include Criollo and Forastero. Forastero is the most widely grown variety worldwide, characterized by weak chocolate flavor, resistance to pests, and thick pulp, with a regular fermentation period of 5–6 days [11]. Criollo has a strong chocolate flavor but is vulnerable to pest attacks, with a faster fermentation period compared to Forastero [12]. Meanwhile, the processing methods in Indonesian cocoa industries are conducted with fermentation, semi-fermentation, and some not fermenting. A previous study revealed that fermented cocoa bean provides better quality compared to non-fermented ones because it produces a stronger chocolate flavor and lower bitterness [13]. The weakness of Indonesian cocoa, as it faces competition in the global market, is the low quality of the bean due to the fact that there are still unfermented beans in several industries [14].

The differences in cocoa varieties and processing methods (fermentation, drying, and roasting) in industries not only result in varied bean qualities but also alter the chemical and bioactive characteristics of the shell [15]. Nonetheless, to date, no study has yet compared the effects of various cocoa varieties, as well as their processing techniques, on the chemical properties of CBS. Therefore, this study aims to identify proximate composition and bioactive compounds in CBS produced at several cocoa industries in Indonesia utilizing different cocoa varieties and processing techniques.

## 2. Materials and Methods

### 2.1. Materials

The chemicals used in the sample analysis were pro-analytical grade chemicals, including hexane (Merck, Darmstadt, Germany), Kjeldahl tablets (Merck, Darmstadt, Germany), H_2_SO_4_ (sulfuric acid) (Merck, Darmstadt, Germany), Na_2_S_2_O_3_ (sodium thiosulfate) (Merck, Darmstadt, Germany), H_3_BO_3_ (boric acid) (Merck, Darmstadt, Germany), methyl red indicator (TCI, Portland, OR, USA), methyl blue (TCI, Portland, OR, USA), phenolphthalein (TCI, Portland, OR, USA), HCl (hydrochloric acid) (Merck, Darmstadt, Germany), lead acetate (Pb-acetate) (Merck Darmstadt Germany), Na_2_HPO_4_ (sodium hydrogen phosphate) (Merck, Darmstadt, Germany), NaOH (sodium hydroxide) (Merck, Darmstadt, Germany), Luff Schoorl solution (Honeywell Fluka, Seelze, Germany), ethanol (Merck, Darmstadt, Germany), Folin–Ciocalteu reagent (Merck, Darmstadt, Germany), Na_2_CO_3_ (sodium carbonate) (Merck, Darmstadt, Germany), methanol (Merck, Darmstadt, Germany), DPPH solution (TCI, Tokyo, Japan), and KI (potassium iodide) (TCI, Portland, OR, USA).

In addition, the materials used in this study were fresh and roasted CBS obtained from several cocoa industries in Indonesia with different varieties and processing methods (Table 1).

Specifically, the fermentation technique in Pasir Ucing, Kendeng Lembu, and Gunung Kidul was performed by spreading cocoa bean in a wooden box. The box was covered with gunny sacks or banana leaves. Every 24 h, the cocoa bean was turned over. After fermentation was complete, the bean was washed using running water to remove the attached pulp, except for in Pasir Ucing and South Sulawesi, followed by drying. In addition, the roasting process of the cocoa bean was performed at 110–120 °C until 1–2% moisture content was reached. Next, the bean was ground, and the shells were separated from the nibs.

### 2.2. Extract Preparations

Extract preparations to analyze total phenolic, theobromine, caffeine, and antioxidant activity were performed based on the method described by Chowdhury et al. [16]. At first, CBS was ground using a grinder, then sieved with a 60-mesh sieve. About 20 g of CBS were then soaked in 96% ethanol solvent at room temperature for 12 h under closed conditions to avoid contamination and light. Extraction was repeated 3 times. Afterward, the filtrate was filtered and evaporated using a rotary evaporator (Buchi Rotavapor R-300, New Castle, DE, USA) to obtain a concentrated extract.

### 2.3. Determination of Proximate Content

Proximate (moisture, ash, fat, protein, and carbohydrate) was analyzed using AOAC methods [17]. Moisture and ash content were determined using the thermogravimetric method. Soxhlet extraction and the Kjedahl method were carried out to calculate fat and protein content, respectively. Meanwhile, carbohydrate was estimated by subtracting moisture, ash, fat, and protein contents from 100%.

### 2.4. Determination of Dietary Fiber Content

The fiber content analysis was evaluated according to AOAC methods [17]. A sample of about 2–4 g was added to 50 mL of 1.25% H_2_SO_4_ and boiled for 30 min using reflux. Next, 50 mL of 3.25% NaOH was mixed with the sample and boiled again for 30 min. The mixture was then hot-filtered using a pre-constantly weighed funnel and filter paper. The residue on the filter paper was washed with 1.25% H_2_SO_4_, hot water, and 96% ethanol. The filter paper was weighed using an analytical balance, dried at 105 °C, and weighed until a constant weight was obtained. If the fiber content was more than 1%, the filter paper and its content were heated in a muffle furnace and weighed until a constant weight is achieved. The formula used to determine the fiber content was as follows:$$\%\;\mathrm{fiber}\;\text{<}\;1\%=\frac{w}{w_{2}}\times 100\%$$
$$\%\;\mathrm{fiber}\;\text{>}\;1\%=\frac{w-w_{1}}{w_{2}}\times 100\%$$

Note:

*w* = mass of sample (g)

*w*_1_ = mass of ash (g)

*w*_2_ = mass of residue on filter paper (g)

### 2.5. Determination of Total Phenolic Content

The phenolic content was determined using the Folin–Ciocalteu method [13]. About 0.3 mL of sample was mixed with 5 mL of 10% Folin–Ciocalteu reagent and 4 mL of 7.5% Na_2_CO_3_. The mixture was incubated at room temperature for 70 min. Absorbance measurement was measured at a wavelength of 750 nm using a UV–Vis spectrophotometer (Rayleigh UV-9200, Beijing, China), and total phenolic content was expressed as mg gallic acid equivalent (GAE) g^−1^.

### 2.6. Determination of Theobromine Content

Theobromine content was evaluated using a High-Performance Liquid Chromatography (HPLC) (LC-20AT, Shimadzu Co., Kyoto, Japan) following the method of Pavia [18] with slight modification. A C18 column (5 μm particle size, i.d. 4.6 × 250 mm) (Shimadzu, Shimadzu Scientific Instruments, Kyoto, Japan) was used in isocratic elution mode with the mobile phase consisting of methanol and water in a ratio of 40:60. The mobile phase and all the solutions were filtered (0.45 μm × 47 mm Millipore nylon filter, Burlington, MA, USA). Meanwhile, the column temperature and the wavelength used were 25 °C and 275 nm, respectively, with a running time of 10 min, and a flow rate of 0.8 mL min^−1^. The extracted sample was dissolved in 10 mL of methanol, sonicated, and injected as 20 μL into the HPLC system. The result was expressed as mg g^−1^ sample.

### 2.7. Determination of Caffeine Content

Determination of caffeine content was performed according to Chowdury et al.’s [16] method with some modification. It was performed using a C18 column (5 μm particle size, i.d. 4.5 × 250 mm) (Shimadzu, Shimadzu Scientific Instruments, Kyoto, Japan) in isocratic elution mode, and the mobile phase consisted of methanol and water in a ratio of 40:60. The detector column was set at 25 °C, with a wavelength of 275 nm and a flow rate of 0.8 mL min^−1^. The extracted sample was dissolved in 10 mL of methanol and then injected into the HPLC system. The caffeine content was expressed as mg g^−1^ sample.

### 2.8. Determination of Antioxidant Activity

Antioxidant activity was analyzed using the 2,2-Diphenyl-1-picrylhydrazyl (DPPH) method [19]. The extracted sample was prepared by diluting the stock solution 40 times to obtain a final concentration of 2.5 µg mL^−1^. Next, 10 µL of the sample was added to 140 µL of DPPH radical solution (250 mM) and the tubes were kept in the dark for 20 min. Absorbance was measured at 536 nm using a UV–Vis spectrophotometer (Rayleigh UV-9200). The DPPH scavenging activity was expressed as a percentage of inhibition using the following formula:
$$\%\;\mathrm{inhibition}=\frac{A\;control-A\;sample}{A\;control}\times 100\%$$

The percentage of inhibition was expressed as IC_50_ in units of concentration (μg mL^−1^) which represents the concentration at which a substance exerts half of its maximal inhibitory effect.

### 2.9. Data Analysis

One-way analysis of variance (ANOVA) was performed to analyze data using Microsoft Excel 2019 (Microsoft, Inc., Redmond, WA, USA) at *p* < 0.05. followed Duncan’s multiple-range test (DMRT).

## 3. Results

### 3.1. Moisture Content

According to Table 2, CBS from each cocoa industry had different moisture contents. The moisture content of fresh CBS was 12.39–16.75% and 6.41–9.23% for roasted CBS. Fresh CBS from Kendeng Lembu-Forastero had the highest moisture content and was significantly different from the other samples. Moreover, all samples from roasted CBS showed a lower moisture content compared to the fresh ones. Roasted CBS from Pasir Ucing-Forastero displayed the highest moisture content among the samples and was also significantly different.

### 3.2. Ash Content

The ash content of CBS varied between 3.74–4.74% for fresh and 6.48–8.21% for roasted form. The Criollo variety had the highest ash content for both fresh and roasted CBS. An increase in ash content from fresh to roasted form was observed, in which Kendeng Lembu-Criollo had the biggest increment and the lowest was from South Sulawesi. Meanwhile, when compared to only Forastero samples, CBS from Pasing Ucing showed the highest increase in ash content (Table 3).

### 3.3. Protein Content

Table 4 represents the protein content of CBS. The protein content of fresh and roasted CBS was 8.20–11.93% and 15.70–18.79%, respectively. Criollo CBS derived from Kendeng Lembu had the greatest protein content among all fresh and roasted shells. Furthermore, all roasted CBS displayed a higher protein compared to a fresh form.

### 3.4. Fat Content

As shown in Table 5, the fat content of CBS was significantly different. The fat content of fresh Criollo CBS was lower than that of Forastero, except for the samples derived from Gunung Kidul. The roasting method caused a reduction in fat content of all CBS samples. The maximum fat content in roasted samples was detected in Gunung Kidul-Forastero, which was statistically similar to Kendeng Lembu-Criollo.

### 3.5. Carbohydrate Content

Table 6 shows that the carbohydrate content of fresh CBS was 34.27–50.75% and 41.04% to 61.36% for roasted CBS. Fresh and roasted Criollo CBS produced the lowest carbohydrate content compared to Forastero. All samples displayed an increment in carbohydrate content after being processed. Roasted CBS from South Sulawesi and Gunung Kidul exhibited the highest carbohydrate compared to other samples.

### 3.6. Dietary Fiber

According to Table 7, the dietary fiber content of CBS varied. Criollo CBS from Kendeng Lembu exhibited the lowest amount of dietary fiber for both fresh as well as roasted samples. Conversely, Gunung Kidul-Forastero had the highest dietary fiber values. In addition, processing caused a slight enhancement in this parameter.

### 3.7. Total Phenolic Content

The phenolic content of CBS is represented in Table 8. The results proved that the phenolic content of fresh CBS was 10.80–35.60 mg GAE g^−1^. Kendeng Lembu-Criollo CBS had the maximum content for fresh samples. After being processed, all samples showed an increase in this parameter. The highest total phenolic content of roasted CBS was also observed in Kendeng Lembu-Criollo and South Sulawesi-Forastero.

### 3.8. Theobromine Content

As shown in Table 9, the theobromine content of fresh CBS was not significantly different between samples, ranging from 7.10–9.60 mg g^−1^. Meanwhile, the greatest theobromine content for roasted CBS was found in Kendeng Lembu-Criollo, but was statistically similar to Kendeng Lembu-Forastero and Gunung Kidul-Forastero. An increase in theobromine was also observed when processing was completed.

### 3.9. Caffeine Content

According to Table 10, all fresh samples exhibited a significant difference in terms of caffeine content. Fresh CBS from Kendeng Lembu-Criollo showed the highest caffeine content. Additionally, processing increased the caffeine content of all samples, but the values were statistically similar, ranging from 3.70 to 6.50 mg g^−1^.

### 3.10. Antioxidant Activity

Table 11 shows that the IC_50_ of fresh CBS was notably different. The lowest IC_50_ was achieved from fresh Criollo CBS from Kendeng-Lembu. A huge drop in IC_50_ was observed after processing was complete. In addition, the lowest value in roasted CBS was also observed in Kendeng Lembu-Criollo, indicating greater antioxidant activity.

## 4. Discussions

Moisture content is a crucial parameter, as CBS have hygroscopic properties and are very prone to fungal growth in high moisture condition [20]. The chemical composition of CBS is affected by climatic conditions as well as varieties [21]. Fresh CBS, in the present study, had a relatively high moisture content, which decreased after processing. According to the Codex Alimentarius Commission [22], the standard moisture content for cocoa bean should be below 7.50%. In this study, roasted CBS from Gunung Kidul-Forastero displayed the lowest moisture content (6.41%). The duration of fermentation influences CBS’s moisture content. The bean from Gunung Kidul had the longest duration of fermentation. A previous study reported that the moisture content of cocoa beans decreases as the time of fermentation time increases [23].

The ash content of CBS reflects the number of inorganic substances [24]. Forastero CBS from South Sulawesi in roasted form showed the lowest ash content compared to other industries. This may be due to no fermentation process being conducted. On the other hand, Forastero CBS from Pasir Ucing showed the biggest increase in ash content after processing. The samples from Pasir Ucing did not undergo pulp washing and this may affect the notable increase in this content. Pulp washing physically removes the ash attached to CBS, which is produced during the fermentation process [25]. The high increase in ash content in this sample was also attributed to the sun-drying method, as this method may contaminate cocoa bean with environmental contaminants [26]. Contaminants from polluted air include particulate matter, such as metals like iron, copper, zinc, nickel, and vanadium [27]. On the other hand, the roasting process also enhanced the ash content of CBS by about 15% [28].

CBS contains a high amount of protein [20]. However, a small portion of this protein consists of D-amino acids, which have lower digestibility compared to L-amino acids. Additionally, the number of D-amino acids increased during the roasting process. In addition, both D-amino acids and L-amino acids contribute to flavor formation [29]. In this study, protein content from fresh CBS varied. The same trend also was found by Assa et al. [30]. From the analysis, it was observed that the roasted Forastero CBS from South Sulawesi showed the lowest protein content due to no fermentation being performed. Protein–polyphenol complexes are formed during the fermentation process [31], and there is a migration of those complexes to the CBS during fermentation [32]. On the other hand, the highest protein content was observed in the Kendeng Lembu-Criollo samples, which was associated with a longer fermentation time. Fermentation led to the accumulation of *Saccharomyces cerevisiae*, which increased protein content [33].

Cocoa fat contributes to providing density and consistency to chocolate, as well as its melting characteristics. It is important to know the amount of fat present in CBS for further proper utilization. The fat content in dried CBS was significantly lower than in the pulp, at approximately 4.09% [34]. When compared to the fat in cocoa nibs, the fat in CBS was more acidic and contained more components that could not be saponified [20]. According to Ostrowska-Ligęza et al. [35], the fat content of Criollo CBS was lower than that of Forastero. The sample from South Sulawesi resulted in the largest decrease in fat content of 4.06% after roasting, likely due to having the highest fat content in fresh samples and the absence of fermentation. Fermentation contributes to higher lipid components in CBS [25]. Furthermore, longer fermentation time also enhances the fat content in CBS [36]. Accordingly, the Gunung Kidul-Forastero CBS, which underwent the longest fermentation period of 5 d, exhibited the lowest decrease in fat content.

The main component of CBS is carbohydrates and approximately 50% of them are fiber [37]. The carbohydrates in CBS consist of sugars, starch, etc. In non-fermented CBS, the most dominant types of sugar are non-reducing sugar and sucrose, which account for around 90% [38]. However, sucrose undergoes a reduction during the fermentation process due to the presence of enzyme invertase, which hydrolyzes sucrose into reducing sugars, such as glucose and fructose [39]. The maximum carbohydrate content was found in the Gunung Kidul-Forastero roasted samples, which had a longer fermentation period of 5 d. Similar results were observed in South Sulawesi-Forastero, but this might be because fresh samples from this industry exhibited the largest concentration of carbohydrates, so the content remained higher after being processed.

CBS consists of a complex lignin–cellulose structure and has a high fiber content ranging from 18.60% to 60.60% [40]. Consumption of fiber is essential for maintaining intestinal health. Fiber is also beneficial for reducing cholesterol and triglyceride levels and lowering the risk of diabetes [20]. The crude fiber in CBS decreased during the fermentation, while increasing during the drying process [41]. The reduction in crude fiber is more significant with longer fermentation due to the activity of probiotic microbes that digest crude fiber [42]. The increase in crude fiber after drying occurs because of the loss of water from the CBS, resulting in a relatively higher content of crude fiber. In the present study, the largest dietary fiber was achieved in Gunung Kidul-Forastero which was associated with mixed drying methods.

Phenolic is a compound found in CBS that has anti-inflammatory and antioxidant properties. It is the primary constituent contributing to antioxidant activity [43]. Cocoa bean processing increased the phenolic content for all samples in our study. The phenolic in Criollo cocoa nibs is about 41.45% higher than in Forastero [44]. Besides the variety, other factors such as climate, sunlight intensity, fruit ripeness, harvesting time, and pest attacks can also affect the phenolic content in cocoa [9]. In addition, the presence or absence of fermentation and the difference in duration also alter this content [45]. Longer fermentation processes reduced phenolic content in CBS [46]. This was in agreement with our study, in which the lowest phenolic content was found in fermented CBS such as Gunung Kidul, Pasir Ucing, and Kendeng Lembu-Forastero samples. During cocoa processing, the drying can also increase the phenolic content in the cocoa beans [47].

Theobromine is an alkaloid compound that acts as a mild stimulant, providing a feeling of excitement and freshness to those who consume it. It also promotes the release of other compounds that can offer a sense of comfort and reduce stress [48]. Fermentation increased the theobromine of CBS due to metabolic processes and migration from nibs [49]. This aligns with the results of this study, where the non-fermented CBS from South Sulawesi exhibited the minimum theobromine content. In addition, among all the samples, roasted Criollo CBS from Kendeng Lembu showed the highest content, although it was statistically similar to Gunung Kidul and Kendeng Lembu-Forastero. Criollo cells are highly permeable to theobromine, so more migration of theobromine occurs during the fermentation process [50].

The caffeine content in cocoa is relatively low, but it is approximately ten times higher than theobromine [51]. Caffeine has beneficial effects on human health, such as preventing Alzheimer’s disease if consumed in recommended amounts [52]. Along with theobromine, caffeine plays a role in the bitter taste of chocolate [53]. In our study, caffeine content in fresh Criollo CBS had a higher value compared to Forastero. Another study also revealed that the caffeine in Criollo CBS was about two times higher than in Forastero [48]. Caffeine is also influenced by cocoa varieties and processing methods. In this study, although statistically there were no differences, processing increased the caffeine levels for all samples. In this regard, the increase observed in non-fermented CBS (South Sulawesi) was higher than that of fermented samples. According to Calvo et al. [54], fermentation could reduce the caffeine content of CBS. They also stated that the drying process reduced the moisture content of CBS but increased the caffeine concentration.

Antioxidants have the ability to neutralize free radicals by preventing oxidation reactions. The compounds that act as antioxidants in CBS are epigallocatechin gallate, quercetin, catechin, epicatechin, apigenin, proanthocyanidins, and gallic acid [55]. Fresh Criollo CBS from Kendeng Lembu had the lowest IC_50_, which means higher antioxidant activity. Fajardo et al. [44] reported that Criollo CBS showed a higher antioxidant activity than Forastero. The lowest values of IC_50_ or the greatest antioxidant activity was observed in Kendeng Lembu-Criollo, which underwent 3 d of fermentation. Fermentation in cocoa beans damages cell permeability, resulting in the diffusion of bioactive compounds from the nibs to the shell, and subsequently higher antioxidant activity of CBS [46].

## 5. Conclusions

The different varieties of cocoa, both Criollo and Forastero, as well as the different processing methods, resulted in varying chemical compositions of cocoa bean shell after roasting. Based on the research findings, roasted cocoa bean shell from Criollo in Kendeng Lembu had greater ash (8.21%) and protein (18.79%) content than other industries. In addition, the concentration of bioactive substances was higher here than it was elsewhere, including total phenolic (136.20 mg GAE g^−1^) and theobromine (22.50 mg g^−1^). Moreover, the lowest ash and protein concentration was observed in CBS made from non-fermented Forastero cocoa pods, like Sulawesi cocoa. The concentration of theobromine (15.40 mg g^−1^) was also lower than in other industries.

## Figures and Tables

**Table 1 foods-12-03316-t001:** Materials and processing methods used in the present study.

Variety	Industry Location	Processing Methods
Forastero	South Sulawesi (4°18′26.3″ S 120°01′22.8″ E)	non-fermented, mechanical and sun drying
Forastero	Pasir Ucing, West Java (6°46′21.3″ S 107°21′13.8″ E)	fermented for 3 d and sun drying
Criollo	Kendeng Lembu, East Java (8°21′51.6″ S 114°01′22.0″ E)	fermented for 3 d, pulp washing, mechanical and sun drying
Forastero	Kendeng Lembu, East Java (8°21′51.6″ S 114°01′22.0″ E)	fermented for 4 d, pulp washing, mechanical and sun drying
Forastero	Gunung Kidul, Central Java (7°50′54.7″ S 110°32′19.2″ E)	fermented for 5 d, pulp washing, mechanical and sun drying

**Table 2 foods-12-03316-t002:** Moisture content (%) of fresh and roasted cocoa bean shell from different cocoa industries.

Sample	Fresh CBS	Roasted CBS
South Sulawesi-Forastero	14.52 ± 0.35 ^b^	7.68 ± 0.35 ^c^
Pasir Ucing-Forastero	12.39 ± 0.28 ^c^	9.23 ± 0.28 ^a^
Kendeng Lembu-Criollo	14.63 ± 0.42 ^b^	7.45 ± 0.02 ^c^
Kendeng Lembu-Forastero	16.75 ± 0.42 ^a^	8.55 ± 0.02 ^b^
Gunung Kidul-Forastero	12.41 ± 0.42 ^c^	6.41 ± 0.02 ^d^

Mean values followed by different superscript letters on the same column are significantly different according to Duncan test at 5% level.

**Table 3 foods-12-03316-t003:** Ash content (%) of fresh and roasted cocoa bean shell from different cocoa industries.

Sample	Fresh CBS	Roasted CBS
South Sulawesi-Forastero	4.39 ± 0.35 ^b^	6.48 ± 0.35 ^c^
Pasir Ucing-Forastero	3.74 ± 0.28 ^c^	7.07 ± 0.28 ^b^
Kendeng Lembu-Criollo	4.74 ± 0.07 ^a^	8.21 ± 0.02 ^a^
Kendeng Lembu-Forastero	4.18 ± 0.07 ^b^	7.23 ± 0.02 ^b^
Gunung Kidul-Forastero	4.32 ± 0.07 ^b^	7.47 ± 0.02 ^b^

Mean values followed by different superscript letters on the same column are significantly different according to Duncan test at 5% level.

**Table 4 foods-12-03316-t004:** Protein content (%) of fresh and roasted cocoa bean shell from different cocoa industries.

Sample	Fresh CBS	Roasted CBS
South Sulawesi-Forastero	10.30 ± 0.35 ^b^	15.70 ± 0.35 ^d^
Pasir Ucing-Forastero	9.90 ± 0.28 ^b^	18.54 ± 0.28 ^b^
Kendeng Lembu-Criollo	11.93 ± 0.14 ^a^	18.79 ± 0.01 ^a^
Kendeng Lembu-Forastero	10.08 ± 0.14 ^b^	17.30 ± 0.01 ^c^
Gunung Kidul-Forastero	8.20 ± 0.14 ^c^	17.15 ± 0.01 ^c^

Mean values followed by different superscript letters on the same column are significantly different according to Duncan test at 5% level.

**Table 5 foods-12-03316-t005:** Fat content (%) of fresh and roasted cocoa bean shell from different cocoa industries.

Sample	Fresh CBS	Roasted CBS
South Sulawesi-Forastero	6.92 ± 0.35 ^a^	2.86 ± 0.35 ^a^
Pasir Ucing-Forastero	3.66 ± 0.28 ^c^	3.12 ± 0.28 ^a^
Kendeng Lembu-Criollo	2.84 ± 0.28 ^d^	2.64 ± 0.03 ^ab^
Kendeng Lembu-Forastero	5.13 ± 0.28 ^b^	3.03 ± 0.03 ^a^
Gunung Kidul-Forastero	2.43 ± 0.28 ^d^	2.28 ± 0.03 ^b^

Mean values followed by different superscript letters on the same column are significantly different according to Duncan test at 5% level.

**Table 6 foods-12-03316-t006:** Carbohydrate content (%) of fresh and roasted cocoa bean shell from different cocoa industries.

Sample	Fresh CBS	Roasted CBS
South Sulawesi-Forastero	50.75 ± 0.35 ^a^	61.15 ± 0.35 ^a^
Pasir Ucing-Forastero	45.91 ± 0.28 ^b^	55.39 ± 0.28 ^c^
Kendeng Lembu-Criollo	34.27 ± 0.35 ^d^	41.04 ± 0.02 ^d^
Kendeng Lembu-Forastero	42.27 ± 0.35 ^c^	60.43 ± 0.02 ^b^
Gunung Kidul-Forastero	43.11 ± 0.35 ^c^	61.36 ± 0.02 ^a^

Mean values followed by different superscript letters on the same column are significantly different according to Duncan test at 5% level.

**Table 7 foods-12-03316-t007:** Dietary fiber (%) of fresh and roasted cocoa bean shell from different cocoa industries.

Sample	Fresh CBS	Roasted CBS
South Sulawesi-Forastero	19.08 ± 0.35 ^b^	19.13 ± 0.35 ^bc^
Pasir Ucing-Forastero	18.33 ± 0.28 ^b^	18.33 ± 0.28 ^c^
Kendeng Lembu-Criollo	15.54 ± 0.04 ^c^	15.60 ± 0.02 ^d^
Kendeng Lembu-Forastero	18.68 ± 0.04 ^b^	18.69 ± 0.02 ^bc^
Gunung Kidul-Forastero	21.50 ± 0.04 ^a^	21.52 ± 0.02 ^a^

Mean values followed by different superscript letters on the same column are significantly different according to Duncan test at 5% level.

**Table 8 foods-12-03316-t008:** Total phenolic content (mg GAE g^−1^) of fresh and roasted cocoa bean shell from different cocoa industries.

Sample	Fresh CBS	Roasted CBS
South Sulawesi-Forastero	24.50 ± 1.54 ^b^	133.10 ± 1.49 ^a^
Pasir Ucing-Forastero	10.80 ± 1.42 ^d^	121.20 ± 0.52 ^b^
Kendeng Lembu-Criollo	35.60 ± 1.35 ^a^	136.20 ± 0.35 ^a^
Kendeng Lembu-Forastero	22.90 ± 0.05 ^c^	115.10 ± 0.50 ^b^
Gunung Kidul-Forastero	22.10 ± 0.05 ^cd^	109.80 ± 0.08 ^b^

Mean values followed by different superscript letters on the same column are significantly different according to Duncan test at 5% level.

**Table 9 foods-12-03316-t009:** Theobromine content (mg g^−1^) of fresh and roasted cocoa bean shell from different cocoa industries.

Sample	Fresh CBS	Roasted CBS
South Sulawesi-Forastero	9.30 ± 0.50 ^a^	15.40 ± 0.40 ^c^
Pasir Ucing-Forastero	7.10 ± 0.21 ^a^	16.50 ± 0.60 ^bc^
Kendeng Lembu-Criollo	9.60 ± 0.14 ^a^	22.50 ± 0.80 ^a^
Kendeng Lembu-Forastero	7.30 ± 0.14 ^a^	19.20 ± 0.10 ^abc^
Gunung Kidul-Forastero	7.80 ± 0.14 ^a^	20.30 ± 0.06 ^ab^

Mean values followed by different superscript letters on the same column are significantly different according to Duncan test at 5% level.

**Table 10 foods-12-03316-t010:** Caffeine content (mg g^−1^) of fresh and roasted cocoa bean shell from different cocoa industries.

Sample	Fresh CBS	Roasted CBS
South Sulawesi-Forastero	2.50 ± 0.71 ^c^	4.90 ± 0.54 ^a^
Pasir Ucing-Forastero	3.50 ± 0.07 ^b^	3.70 ± 0.07 ^a^
Kendeng Lembu-Criollo	6.30 ± 0.14 ^a^	6.50 ± 0.07 ^a^
Kendeng Lembu-Forastero	3.80 ± 0.14 ^b^	4.80 ± 0.07 ^a^
Gunung Kidul-Forastero	3.20 ± 0.14 ^bc^	3.70 ± 0.07 ^a^

Mean values followed by different superscript letters on the same column are significantly different according to Duncan test at 5% level.

**Table 11 foods-12-03316-t011:** IC_50_ (µg mL^−1^) of fresh and roasted cocoa bean shell from different cocoa industries.

Sample	Fresh CBS	Roasted CBS
South Sulawesi-Forastero	190.05 ± 0.35 ^d^	102.85 ± 0.35 ^b^
Pasir Ucing-Forastero	197.93 ± 0.28 ^b^	55.20 ± 0.28 ^c^
Kendeng Lembu-Criollo	170.47 ± 0.03 ^e^	49.79 ± 0.05 ^d^
Kendeng Lembu-Forastero	194.96 ± 0.03 ^c^	105.37 ± 0.05 ^a^
Gunung Kidul-Forastero	199.96 ± 0.03 ^a^	55.37 ± 0.05 ^c^

Mean values followed by different superscript letters on the same column are significantly different according to Duncan test at 5% level.

## Data Availability

The data used to support the findings of this study can be made available by the corresponding author upon request.

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
