# Peer review of "Proximate Composition and Bioactive Compounds of Cocoa Bean Shells as a By-Product from Cocoa Industries in Indonesia"

_foods, 2023, doi:10.3390/foods12173316_

Round 1

Reviewer 1 Report

The manuscript entitled "Proximate and Bioactive Compounds of Cocoa Bean Shells as a By-Product from Cocoa Industries in Indonesia" is a good work by the authors. However, there needs extensive correction in the manuscript. 

1. The title needs to be modified as Proximate composition and Bioactive compounds 

2. Abstract is vague and doesn't contain any quantitative data. I recommend to re-write the abstract by incorporating quantitative data

3. The introduction is poor with limited literature support. Rewrite the section by incorporating the global status of Cocoa production, its utility, health benefits, range of compounds and previous reports on Proximate composition.

4. Geographical coordinates may be included for collection sites

5. The DPPH assay alone cannot indicate the antioxidant potentials. Include additional experiments 

6. Figures lacks standard deviation or SE

7. Did the authors identified IC50 for DPPH assay? If so mention methods and clearly indicate the results

8. The use of figures seems to be improper; many of these data can be presented as tables

9. There are plenty of typographic errors and language corrections. A thorough proof reading is refommended

There are plenty of typographic errors and language corrections. A thorough proof reading is refommended

Reviewer 2 Report

Dear authors,

Correct the manuscript based on the following comments.

Abstract: It is superficial and does not report the findings. After a brief contextualization, write what was done to valorize cocoa by-products and the main results. If applicable write the novelty of this research and for what the results would serve. Also, how different is the Indonesian cocoa from somewhere else's?

L39 - Theobromine is not a lipid. It is an alkaloid.

L51-56 - There are many documents reporting the effects of multiple processes on the quality of cocoa byproducts. Which type of ''processing'' are the authors talking about? What is the current application of shells, and what is the importance to study that? Are the shells considered waste? The novelty is still not clear.

L-78 - ''..by 10'' - Fix the mistake by mentioning the authors before the reference.

L-65-67 - The authors must mention from which companies/location the byproducts were donated. Include the conditions used for each company for fermentation/roasting (if possible)

L77-80 - Rewrite the topic. How the raw material was milled?  For how long the extractions were performed? Were the extractions done in dark, under mixing? How many replicates were done?

L85-88 - Rewrite the last part, mentioning that carbohydrates were estimated by subtraction.

Topics 2.6 and 2.7- include column particle size, brand, city, country. Were the samples filtered before injected? Which type of filter was used? Was the oven running at room temperature? Was the elution done isocratically?

Topic 2.8 - Replace ''activity'' per ''potential''. The authors are measuring the potential of sample to scavenge the free radical, and not activity.

Topic 2.9 - Which program was used? Which type of correlation was used? The authors also must mention which type of statistical test was done to detect significant differences among samples.

For figures and graphics use arial or calibri; include the statistical differences as well.

Figure 7 - Normally the polyphenol measured by FC reagent is expressed as mg gallic acid eq. How this % was calculated?

Figure 10 - There is a mistake in here. In MM, the authors mentioned that antioxidant capacity would be measured in terms of % inhibition and here the results were expressed as ug/ml.

L362-363 - The fermentation was not studied in here; the authors only compared ''fresh'' and ''roasted'' samples. Also, the fermentation conditions performed by each company from where the samples were donated were not provided.

Round 2

Reviewer 1 Report

No more comments

Author Response

Thank you very much

Reviewer 2 Report

Dear authors,

The quality of manuscript improved. There are additional comments for authors consider:

Topic 2.1 - Include brand, city, country of chemicals used.

L153 - I do not think it is needed include the ''x100%'' once the results were expressed in terms of 100g raw material

Topics 2.6 and 2.7 - Results were expressed in terms of what? mg/100g eraw material?

Tables 2-11 - Since the authors are comparing processes, the statistical difference was supposed to be done across the lines for each variety (crude x fermented)

l443-445 - Polyphenols consist of a class of compounds. What do authors mean? Are you correlating the total phenolic content measured with FC reagent with antioxidant capacity? If yes, rewrite the sentence

l446 - replace ''polyphenol level'' to: total phenolic content estimated. Do this throughout the text to avoid misinterpretations. Also, the authors did not measure any individual polyphenol probably present in samples.

l483-l484 - These phenolic substances were not measured. What do you mean?

L488 - Antioxidant content or inhibition percentage (IC50)?

l185- Table 11 - Is the inhibition percentage (IC50) expressed as ''%'' or as concentration? Correct this.

Compounds measured by HPLC - Instead of percentage, express the results in terms of concentration (mass compound detected per mass of raw material). The results as % sounds confusing (% of compounds present in extract or present in raw material?)
